# The Role of Agroforestry in Poverty Alleviation: A Case Study from Nujiang Prefecture, Southwestern China

Yaquan Dou [1] , Ya Li [2], Ming Li [1], Xingliang Chen [1] and Xiaodi Zhao [1,3,*]

1 Research Institute of Forestry Policy and Information, Chinese Academy of Forestry, Beijing 100091, China; douyq@caf.ac.cn (Y.D.); ming_li@caf.ac.cn (M.L.); chen62889299@126.com (X.C.)
2 College of Economics and Management, Southwest Forestry University, Kunming 650224, China; liy@swfu.edu.cn
3 Faculty of Forestry, The University of British Columbia, Vancouver, BC V6T 1Z4, Canada
* Correspondence: zhaoxiaodi@caf.ac.cn; Tel.: +86-136-7105-3260

**Abstract:** Agroforestry has gained increasing attention as a sustainable land use mode to ensure food security, mitigate global climate change, and improve farmers' livelihoods. Likewise, agroforestry plays a key role in alleviating poverty, mitigating climate change and achieving the Sustainable Development Goals (SDGs) in China. *Lanxangia tsaoko*, as a typical agroforestry species in Nujiang Prefecture, plays a vital role in improving farmers' livelihoods. After years of development, the *Lanxangia tsaoko* industry (LTI) in Nujiang Prefecture has made remarkable achievements and accumulated useful experiences. Taking the development of LTI as an example, this paper analyzes the impact of agroforestry on farmers' livelihoods and its mechanism through field survey and theoretical deduction. First, by investigating the willingness of households to choose LTI, we found that most farmers have a positive attitude toward LTI, and they actively participate in cooperative organizations. Then, by analyzing the development stage and mode of LTI in Nujiang Prefecture, we found that the industry has made significant progress under the external and internal effects. We also found that the mechanism by which agroforestry affects farmers' livelihoods is a process in which various stakeholders, including government, enterprises, farmers, etc., participate in industrial development with their production components and advantages. Although agroforestry is an effective way to improve farmers' livelihoods and promote sustainable agricultural development, it is also necessary to pay attention to risk prevention. This paper discusses the role of agroforestry in farmers' livelihoods, which provides a reference for lower-income forest areas.

**Keywords:** agroforestry; sustainable forestry development; Case Study Method; practical experiences; implications





## 1. Introduction

As COVID-19 and climate change continuously impact the global food production and supply system, the global food security situation is deteriorating [1–3]. Therefore, in order to ensure food security, the world is looking for a viable path to sustainable agricultural development [4]. As main components of terrestrial ecosystems, forests play a key role in regulating the global balance, slowing the increase of greenhouse gas concentrations such as atmospheric $CO_2$ [5]. Some international climate commitments and obligations such as the United Nations Framework Convention on Climate Change (UNFCCC), the Paris Agreement (PA), and the Glasgow Climate Pact (GCP), have put the protection of global forest resources and the sustainable development of forestry under a severe test [6]. Agroforestry can effectively protect forest resources, cope with climate change, ensure food security, and achieve sustainable development [7–11]. Therefore, agroforestry, as a multifunctional and environment-friendly land use mode, is attracting more and more global attention [12–14]. In particular, since the signing of the "Kyoto Protocol", agroforestry has gained increased attention as a strategy to ensure food security and mitigate global climate

change [15–17]. For example, Kay et al. (2019) found that agroforestry in Priority Areas could lead to a sequestration of 2.1 to 63.9 million t C a$^{-1}$ (7.78 and 234.85 million t $CO_2$eq a$^{-1}$) depending on the type of agroforestry. This corresponds to between 1.4 and 43.4% of European agricultural greenhouse gas (GHG) emissions [5]. Meanwhile, collecting soil samples from teff–Acacia agroforestry and conventional teff fields at two different sites, Kim et al. (2022) demonstrated that locally adopted agroforestry practices can increase soil organic carbon and nutrients in the long term, thereby contributing to enhanced soil fertility and improved climate change mitigation strategies via carbon sequestration in northwestern Ethiopia [18]. Ballesteros-Possú et al. (2022) found that cacao agroforestry arrangements increased cacao yield and carbon storage, becoming a suitable alternative to improving traditional systems [19]. In addition, agroforestry plays a key role in improving the livelihood and development capacity of farmers [20–23]. For instance, Akter et al. (2022) used a mixed-method strategy that included a survey, focus group discussion, key informant interviews, and direct observation from 150 tribal farmers practicing different types of agroforestry systems in Tangail, Bangladesh. They found that agroforestry systems have provided numerous benefits and greatly enhanced farmers' livelihoods through better access to food, timber, fodder, and fuelwood and greater access to livelihood capitals (except social capital) [24]. De Giusti et al. (2019) found that agroforestry plays an important economic and environmental role by supporting subsistence through provision of fuelwood and could relieve pressure upon common forest resources in Western Kenya [25]. Likewise, using a log linear regression model with cross-sectional data collected from a sample of 300 households, Zerihun (2021) explored the likely impact of agroforestry practices on promoting the livelihood of rural communities in the study areas and found that average household income increased as a function of utilization of agroforestry practices in South Africa [26]. Additionally, Wijayanto et al. (2022) found that agroforestry adoption had a significant and positive impact on subjective well-being indicators using propensity score matching (PSM). Farmers who adopted agroforestry were happier and more satisfied than those who did not adopt agroforestry in East Java, Indonesia [27].

On 22 February 2022, the Communist Party of China (CPC) Central Committee and the General Office of the State Council of China, issued the "No. 1 central document" for 2022, which clearly stated that "we should firmly stick to the two bottom lines of ensuring national food security and preventing large-scale poverty" [28]. For decades, forest resources have provided important strategic resources for economic and social development, safeguarded national security such as ecological security and food security, and played a key role in improving the livelihood of farmers in China [29]. For example, forests are "grain depots", which can provide humans with fruits, seeds, nuts, rhizomes, tubers, fungi, and other kinds of food, so as to effectively ensure national food security. Also, forests are "money depots", which can continuously provide humans with a variety of products, including wood, energy materials, animal and plant by-products, and chemical and pharmaceutical resources [30,31]. However, with sharp decreases in forest resources and increasingly prominent global environmental problems, the contradiction between economic development and forest resource protection is becoming increasingly serious [32]. In response to this problem, China has successively implemented major ecological construction projects such as natural forest resource protection, returning farmland to forests and grasslands, and forest ecological public welfare projects [33,34]. In particular, since the 18th National Congress of the Communist Party of China, China has attached great importance to the "construction of ecological civilization", and put forward a series of important statements, such as "Lucid waters and lush mountains are invaluable assets" [35–37]. Against this background, forestry has gradually transformed from timber-based management to multi-purpose forest management, and to sustainable forestry management in China [38–40]. It has been proven that agroforestry can make up for the shortcomings of long cycles and slow incomes of forestry in China [41,42]. In addition, agroforestry can realize economic development while protecting forest resources [43,44]. For instance, Dou et al. (2023) constructed a performance evaluation index system to determine the poverty alleviation performance of

the agroforestry industry in Yunnan Province using the analytic hierarchy process (AHP), and they found that the development of the agroforestry industry significantly improved farmers' livelihoods and ecological environments [45]. Moreover, because it effectively conforms to the national conditions of "more population and less land", and well reflects the characteristics of ecological agriculture as "modern, efficient, and circular", agroforestry has become an effective system for the development of ecological agriculture in China [42].

Generally, ecological benefits, such as carbon sequestration capacity, water conservation, and soil improvement, as well as economic benefits, such as farmers' income from agroforestry systems, have been recognized by scholars [46–51], yet the mechanism of agroforestry in improving households' livelihoods is unclear. Against the background of global climate change and food security crisis, it is necessary to explore that the sustainable development path of agroforestry and its impact on livelihoods. There have been few studies on the ways in which agroforestry improves livelihoods, or how agroforestry development affects farmers. In order to examine these issues, this paper takes the development of the *Lanxangia tsaoko* industry (LTI) in Nujiang Prefecture as a typical case to analyze the impact of agroforestry on farmers' livelihoods using empirical research methods such as field observation, in-depth interviews, and a questionnaire survey. First, we use a literature review to clarify the current situation of the development of LTI in Nujiang Prefecture. Second, using the methods of questionnaires, field observation and in-depth interviews, we analyze the development model of LTI and the effect and influence mechanism of improving farmers' livelihoods. Finally, combined with examples to prove the theory, we summarize the general rules of the development of the agroforestry industry from the experience of LTI in Nujiang Prefecture.

## 2. Materials and Methods

### 2.1. Study Area

Nujiang Prefecture is located on the southwest border of China (Figure 1), which includes greatly impoverished areas and is the main battleground for poverty alleviation nationwide of the Aid-the-Poor Development Office of the State Council, China [52]. It has jurisdiction over one county-level city, one county, and two autonomous counties: Lushui City, Fugong County, Gongshan Dulong and Nu Autonomous County, and Lanping Bai and Pumi Autonomous County (Table 1). Its infrastructure is weak, and industrial development is relatively lagging behind, and thus it is difficult to consolidate the achievements of alleviating poverty and promoting rural revitalization [53]. Nujiang Prefecture is located in the core area of the "Three Parallel Rivers" World Natural Heritage site and is one of the regions with the richest biodiversity in the world, which is the reason why it is an important ecological functional area in China. According to statistics, the percentage of forest cover is 78.09% in Nujiang Prefecture, which ranks second in Yunnan Province. In an attempt to solve the problem of poverty and limited development resources, the government has been exploring in practice how to make full use of forestry resources to find a development path for the local population. *Lanxangia tsaoko* is a common spice and an important medicinal plant and thus is a type of food with edible and medicinal value. However, *Lanxangia tsaoko* planting has high requirements for the natural environment. It prefers a warm and humid climate, and it is fragile in hot, dry, or frosty environments. Therefore, it is a typical agroforestry system at altitudes of 1000–2000 m, with an average annual temperature of 15–20 °C, and under the forest canopy with about 50–60% shade. Nujiang Prefecture has superior natural conditions, with the distribution of wild *Lanxangia tsaoko* resources, rich forest resources, abundant rainfall, and suitable soil types for the development of LTI. LTI has experienced the following stages in Nujiang Prefecture.

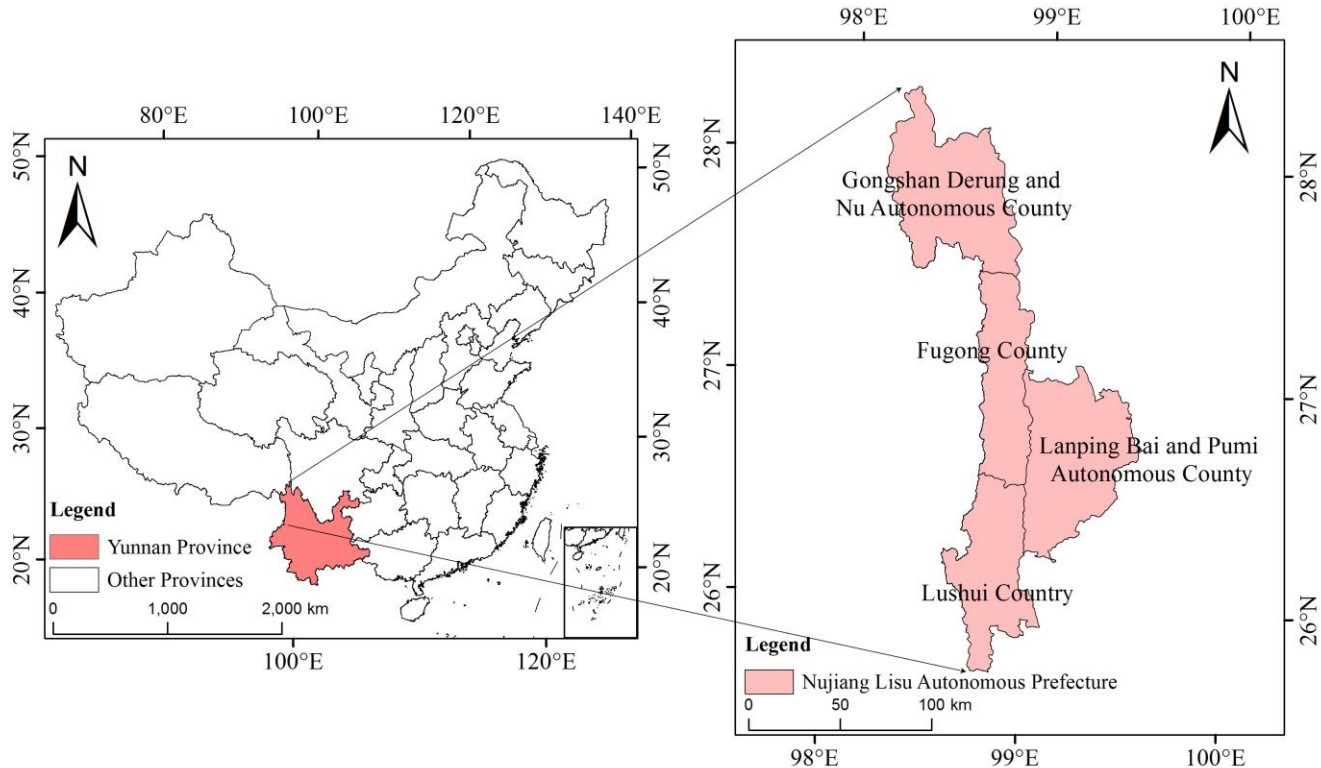

**Figure 1.** Location of the study area.

**Table 1.** Administrative division of Nujiang Prefecture.

| City/Country | Land Area/hm$^2$ | Population | GDP/CNY |
|---|---|---|---|
| Lushui City | 293,800 | $2.03 \times 10^5$ | $8.73 \times 10^9$ |
| Fugong County | 280,400 | $1.14 \times 10^5$ | $2.79 \times 10^9$ |
| Gongshan Dulong and Nu Autonomous County | 445,000 | $1.96 \times 10^5$ | $1.88 \times 10^9$ |
| Lanping Bai and Pumi Autonomous County | 450,600 | $0.38 \times 10^5$ | $9.83 \times 10^9$ |

(1) The embryonic stage: solving food and clothing problems (1978–1994). In the initial stages for reform and opening-up, it was an important task for rural development to solve the problem of meeting farmers' food and clothing needs in China. The government introduced 0.16 hm$^2$ of *Lanxangia tsaoko* and mobilized extensively in 1978. However, given the situation of a strong tradition of thought, low autonomy of individual operation, and high resource constraints, farmers' perceptions of *Lanxangia tsaoko* were inadequate, and they were not satisfied with the statement that "planting *Lanxangia tsaoko* can make people rich". When the government distributed seedlings free of charge and mobilized farmers to participate in planting, there was a phenomenon in which most farmers showed low enthusiasm and had a pessimistic outlook. In fact, farmers regarded this plant as food rather than as a cash crop, resulting in small and scattered planting areas of *Lanxangia tsaoko*, while no effective value was realized.

(2) The growth stage: helping development (1995–2011). With the promotion of a series of important national projects, such as poverty alleviation, agricultural industrialization development, and new rural construction, the government had taken advantage of the situation to promote the development of LTI. On the one hand, the government built a "Green *Lanxangia tsaoko* Industrial Zone" from south to north by utilizing the resources in poor areas. On the other hand, the government actively guided farmers in large-scale operations. Therefore, the planting area of *Lanxangia tsaoko* increased from 0.16 hm$^2$ in 1978 to 23,267 hm$^2$ in 2010, with a total output value of CNY $7.1 \times 10^7$. During this period,

LTI began to play a role in economic development, and served as an ecological way to increase farmers' sustainable income.

(3) The maturity stage: achieving targeted poverty alleviation (2012–2020). From 2012 to 2020, the government proposed the development path of "ecological statehood, green and enriching the people" in Nujiang Prefecture. Through the implementation of the project of returning farmland to forests and the reform of a collective forest right system, farmers actively developed LTI, which had achieved multiple goals of economic development, ecological protection, and social stability. Additionally, in order to meet the market demand, the government had established an Industrial Development Research Institute, which had improved industrial quality and increased farmers' incomes. According to the investigation, there was a planting area of 72,140 hm$^2$ and output value of CNY $2.51 \times 10^8$ in Nujiang Prefecture by 2019. By then, it had become the core production area of *Lanxangia tsaoko* in China and the largest *Lanxangia tsaoko* planting area in Yunnan Province. Many farmers left poverty by developing LTI in Nujiang Prefecture. It was reported that Nujiang Prefecture had built a 7.2 million hm$^2$ national *Lanxangia tsaoko* core production, with a planting scale, with an output value of CNY 500 million, and realizing the poverty alleviation of 40,000 people.

(4) The high-quality stage: promoting rural revitalization (2021–). With the support of the government and driven by enterprises, the government has continuously promoted the transformation of the LTI to become ecological, large-scale, branded, and specialized. Moreover, it is a positive transformation in which farmers gradually realize that the superior quality of forestry resources can support high-quality *Lanxangia tsaoko*. Additionally, they also realize that the development of *Lanxangia tsaoko* planting can not only increase income, but also protect forests. Therefore, in order to expand the planting scale of *Lanxangia tsaoko* and improve its yield and quality, they actively planted evergreen broad-leaf forests to protect and repair the local ecology. The most recent figures show that there were 74,433 hm$^2$ of *Lanxangia tsaoko* and CNY $1.32 \times 10^9$ of output value in Nujiang Prefecture by 2021 [45].

### 2.2. Data Sources

This paper takes Lushui City, Fugong County, and Gongshan Dulong and Nu Autonomous County in Nujiang Prefecture as a typical case (Table 2), and especially takes Lushui City as an important region, to study the impact and mechanism of LTI on farmers' livelihoods. One type of data deals with the situation of LTI. This type of data comes from local government statistics and forestry-related departments. Another type of data regards the willingness of farmers regarding management, which is mainly obtained through questionnaires. The investigation can be divided into the following stages.

**Table 2.** Basic information of the sampled regions.

| City/Country | Forest Area /hm$^2$ | Forest Coverage /% | *Lanxangia tsaoko* Planting Area/hm$^2$ | Proportion of *Lanxangia tsaoko* Planting/% |
|---|---|---|---|---|
| Lushui City | 244,165.5 | 78.98 | 22,207.33 | 29.84 |
| Fugong County | 250,586.2 | 82.23 | 37,333.33 | 50.16 |
| Gongshan Dulong and Nu Autonomous County | 380,933.3 | 83.89 | 14,893.33 | 20.00 |

First, according to the situation of LTI in Nujiang Prefecture, three regions, including Lushui City, Fugong County, and Gongshan Dulong and Nu Autonomous County, were selected as sample cities (counties). Five sample villages were randomly selected from each sample area, for a total of 15 sample villages, to understand the development of LTI and the farmers' livelihoods.

Second, according to the method of random sampling, a proportion of farmers was selected from each sample village to participate in the questionnaire. Through the questionnaire, we can clarify the willingness and behavior of farmers' LTI development.

Third, we selected the relevant heads of two to three counties to conduct in-depth interviews, and we visited LTI bases or enterprises to analyze the development mode of LTI and the mechanism of impact on farmers' livelihoods.

Fourth, we had a discussion with the relevant heads of Nujiang Rural Agriculture Bureau and the Forestry and Grassland Bureau to understand the development situation of LTI in Nujiang Prefecture. Through the above methods, we obtained first-hand information about the development of LTI and farmers' livelihoods.

*2.3. Research Methods*

2.3.1. Case Study Method

The case study method (CSM) consists of combining questions, taking typical cases as the material, and summarizing the general rules through specific analysis. It includes collecting and recording one or several case materials. The CSM is mainly divided into the following steps:

(1) Posing research questions

The main question of this study regards the role of agroforestry in farmers' livelihood. The specific questions are: (1) Will the development of agroforestry improve the livelihood of farmers? How much improvement is it? (2) What is the development mechanism of agroforestry? (3) How does LTI development improve farmers' livelihoods? What is its mechanism?

(2) Developing research hypotheses

Based on the literature and practical experience, this paper proposes the following hypotheses.

**Hypothesis 1:** *LTI has significantly improved farmers' livelihoods.*

**Hypothesis 2:** *External forces and internal motives are important factors for LTI development.*

**Hypothesis 3:** *Stakeholders participate in LTI to jointly promote the improvement of farmers' livelihoods.*

(3) Selecting research cases

The criteria for selecting one or more cases are related to the research objectives and the questions to be answered. For example, a single case study can be used to confirm a theory. The characteristic of a multiple-case study is that it includes single case analyses and their comparison analyses. Additionally, a single case study is a comprehensive analysis of each case as an independent whole, while a multiple-case study is a unified abstraction and induction of all cases.

(4) Collecting case data

There are four main ways to select cases: (1) Documents. (2) Archival records. Different from documents, the usefulness of these archives will vary with different case studies. (3) Interviews. A common method is the focus interview, which is a way to interview respondents in a short period of time. Another type is the questionnaire survey, which is limited to more structured questions. (4) Field observation. In this way, the researcher can visit the case on location.

(5) Analyzing case data

Before analyzing data, researchers need to determine analysis methods. Generally, there are two main methods to analyze cases. One is to explain the case by theory. This method can be based on the question posed at the beginning, and the question reflects new ideas and the results of the previous review. Another is to describe the case. This method conducts case studies by constructing a description framework.

This paper analyzes the role of agroforestry in maintaining farmers' livelihoods by the CSM. First, we analyze the achievement of agroforestry through data collection and a questionnaire to verify Hypothesis 1. Second, we summarize the development stage of agroforestry through field interviews to verify Hypothesis 2. Finally, we analyze the development model of agroforestry through field observation and in-depth interview to verify Hypothesis 3. In addition, we analyze the constraints on the development of agroforestry (Figure 2).

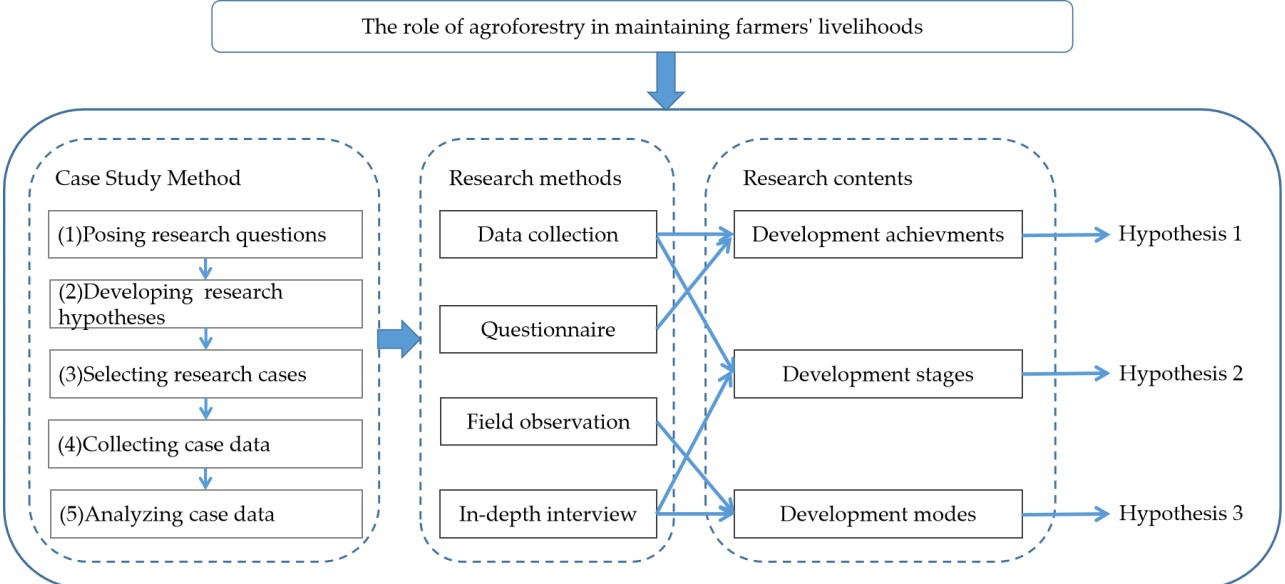

**Figure 2.** Research framework of this paper.

2.3.2. Descriptive Statistical Analysis

Descriptive statistical analysis is a method that uses tabulation and classification, graphs, and calculation of generalized data to describe the characteristics of data. It is a statistical description of all variables in the survey, including frequency analysis, central trend analysis, dispersion degree analysis, and some basic statistical graphs. Frequency analysis is used to calculate the selection frequency and proportion of certain types of data, such as gender. It is used for the statistics of basic background information of samples, as well as the analysis of sample characteristics and basic attitudes. In this paper, we use SPSS24.0 to statistically analyze the questionnaire results by drawing frequency distribution tables. The main contents are as follows: (1) The willingness of the surveyed farmers to choose LTI (Appendix A Table A2). (2) The willingness of the surveyed farmers to choose organizational form. (3) The willingness of the surveyed farmers to choose training.

### 3. Results

*3.1. Basic Information of the Households*

In the survey, a total of 150 questionnaires were collected. After eliminating the missing values of relevant data, 139 valid samples were obtained. As shown in Table 3, the proportions of male and female respondents were 53.2% and 46.8%, respectively. The majority of the respondents were 41–60 years old, accounting for 31.8%, while the smallest group was 31–40 years old. In terms of the nationality of those surveyed, more than half of households are Lieu nationality. The educational levels of households were generally low. Households with an intermediate level of education (Grades 7–9) accounted for 59%. Less than 9.4% of households had a high level education. In addition, 97.8% of the surveyed households bought medical insurance, while 89.2% of them bought endowment insurance. By analyzing the reliability and validity of the questionnaire, we found $\alpha = 0.826 > 0.7$, which shows that the reliability of the questionnaire is very good (Table 4). As shown in

Table 5, the value of the Kaiser–Meyer–Olkin (KMO) statistic is 0.876 > 0.8, which indicates that the questionnaire data are very effective.

**Table 3.** Basic information of the surveyed households.

| Characteristic | Category | Percentage |
|---|---|---|
| Gender | Male | 53.2% |
| | Female | 46.8% |
| Age | 16–30 years old | 21.6% |
| | 31–40 years old | 16.0% |
| | 41–60 years old | 31.8% |
| | Over 60 years old | 30.6% |
| Nationality | Han | 39.6% |
| | Lieu | 58.3% |
| | Other | 2.1% |
| Education | Not attending school | 4.6% |
| | Primary (Grade 1–6) | 25.6% |
| | Middle (Grade 7–9) | 59.0% |
| | High (Grade 10–) | 9.4% |
| Social security | Medical insurance | 97.8% |
| | Endowment insurance | 89.2% |

**Table 4.** Cronbach Reliability Analysis.

| Number of Items | Sample Size | Cronbach $\alpha$ |
|---|---|---|
| 23 | 139 | 0.826 |

**Table 5.** KMO and Bartlett's test.

| KMO | | 0.876 |
|---|---|---|
| Bartlett's sphericity test | Approximate chi square | 1320.467 |
| | df | 55 |
| | $p$ | 0.000 |

*3.2. Willingness of LTI Development*

3.2.1. Most Farmers Have a Positive Attitude about LTI

As listed in Table 6, 79.6% of farmers were willing to expand the scale of the LTI, and 38.7% of them were very optimistic, while 20.4% of farmers were still unwilling, and 22.5% of farmers were quite pessimistic or very pessimistic, which shows that there are certain challenges in the development of LTI. Additionally, it is evident that farmers still face many difficulties, such as lack of capital (28.4%), technology (23.2%), labor (17.9%), sales (14.7%), infrastructure (6.3%), natural resources (3.2%), and suitable projects (6.3%). Therefore, farmers prefer to obtain some support in terms of capital (36.8%) and technology (25.3%). Additionally, they also wish to receive talent support and information support. As for the form of industrial organization, most farmers (32.75%) believe that the most effective is by themselves, while some farmers (23.1%) believe that enterprises can promote industrial development.

**Table 6.** Willingness of the surveyed farmers to choose LTI.

| Question | Category | Percentage |
|---|---|---|
| Are you willing to expand the industry scale ? | Yes | 79.6% |
| | No | 20.4% |
| The attitude of industrial development | Very optimistic | 38.7% |
| | Quite optimistic | 18.4% |
| | Average | 20.4% |

**Table 6.** *Cont.*

| Question | Category | Percentage |
|---|---|---|
| The problem of industrial development | Quite pessimistic | 14.3% |
| | Very pessimistic | 8.2% |
| | Lack of capital | 28.4% |
| | Lack of technology | 23.2% |
| | Lack of labor | 17.9% |
| | Lack of sales channels | 14.7% |
| | Lack of infrastructure | 6.3% |
| | Lack of resources | 3.2% |
| | Lack of projects | 6.3% |
| What kind of support do you want to receive? | Financial support | 36.8% |
| | Technical support | 25.3% |
| | Talent support | 13.8% |
| | Market support | 6.9% |
| | Information support | 10.3% |
| | Policy support | 6.9% |
| The form of industrial organization | Farmers themselves | 32.7% |
| | Enterprises | 23.1% |
| | Village collectives | 9.6% |
| | Cooperatives | 19.2% |
| | Government | 15.4% |

### 3.2.2. Most Farmers Actively Participate in Cooperative Organizations

As shown in Table 7, 32.6% of farmers think that there are no or unclear cooperative organizations. It is obvious that services provided by cooperatives for farmers mainly focus on technological services (27.45%), production services (16.7%), and loan services (16.7%), which have helped farmers develop LTI. However, 14.2% of the farmers consider that cooperatives did not provide specific services. When asked what kind of service they want to receive, most farmers chose technical guidance (29.4%) and financial support (28.4%), which are also the biggest problems faced by farmers in developing LTI. On the whole, most farmers (48.9%) who received support provided by cooperatives were satisfied.

**Table 7.** Willingness of the surveyed farmers to choose organizational form.

| Question | Category | Percentage | Question | Category | Percentage |
|---|---|---|---|---|---|
| Whether you joined cooperatives | Yes | 67.4% | Whether you joined enterprises | Yes | 8.2% |
| | No | 22.4% | | No | 36.7% |
| | Unclear | 10.2% | | Unclear | 55.1% |
| What services have you received from cooperatives? | Production services | 16.7% | What support have you received from enterprises? | Financial support | 14.3% |
| | Technology services | 27.4% | | Technical support | 18.4% |
| | Transportation and marketing services | 11.9% | | Talent support | 8.2% |
| | Processing services | 13.1% | | Market support | 20.4% |
| | Loan services | 16.7% | | Information support | 14.3% |
| | No specific services | 14.2% | | No specific support | 24.4% |
| What kind of service do you want to get? | Financial services | 28.4% | What kind of support do you want to get? | Financial support | 30.6% |
| | Technical guidance | 29.4% | | Technical guidance | 18.4% |
| | Product sales | 9.8% | | Product sales | 14.3% |
| | Talent training | 11.8% | | Talent training | 8.1% |
| | Market support | 12.7% | | Market support | 14.3% |
| | Information services | 7.9% | | Information services | 14.3% |
| Your satisfaction with cooperatives | Very satisfied | 26.5% | Management mode of enterprises cooperation | Enterprises + farmers | 38.8% |
| | Quite satisfied | 22.4% | | Enterprises + bases + farmers | 6.1% |
| | Average | 34.7% | | Enterprises + cooperatives + farmers | 18.4% |
| | Quite unsatisfied | 12.2% | | Others | 36.7% |
| | Very unsatisfied | 4.2% | | | |

The results show that 36.7% of farmers think that there are no enterprises to help them, and more than half (55.1%) do not know whether there is an enterprise. There is a positive discovery that some enterprises have also given some support to farmers, such as expanding the market (20.4%), technical support (18.4%), and financial support (14.3%), and farmers would prefer to receive financial support (30.6%) and technical guidance (18.4%), which proves that capital and technology are the biggest problems in the process of LTI. In the existing cooperative operation mode, most enterprises adopt the "enterprise + farmer" (38.8%) approach. Additionally, some enterprises (36.7%) help farmers to develop LTI by establishing cooperation between the government and farmers.

### 3.2.3. Most Farmers Have a Strong Desire for Skill Training

Results show that most farmers (77.6%) were willing to receive training, while 22.4% of them were unwilling to participate for some reason. A total of 59.2% of farmers have received training on *Lanxangia tsaoko* planting, collection, and other skills. There is a positive discovery that most farmers who have received training are generally satisfied. However, there are still a few farmers who are not satisfied with the training results because of the training content, methods, or other reasons. As shown in Table 8, most farmers (42.2%) would choose to participate in the training according to their own emphasis, while 21.9% were content with their training. Additionally, the training time (7.8%), place (9.4%), method (6.3%), and effect (7.8%) will also affect the enthusiasm of farmers to participate in training. Farmers preferred technical training (41.8%), whether from the government, enterprises, or others. Moreover, some farmers also hoped to receive training on management (20.9%), market development (17.9%), and sales channels and methods (14.9%) about LTI.

**Table 8.** Willingness of the surveyed farmers to choose training.

| Question | Category | Percentage |
|---|---|---|
| Are you willing to participate in training? | Yes | 77.6% |
| | No | 22.4% |
| Have you participated in training? | Yes | 59.2% |
| | No | 40.8% |
| Please evaluate the training effect | Very satisfied | 42.9% |
| | Quite satisfied | 30.6% |
| | Average | 16.3% |
| | Quite unsatisfied | 8.2% |
| | Very unsatisfied | 2.0% |
| Factors influencing farmers' participation in training | Personal concern | 42.2% |
| | Training content | 21.9% |
| | Training methods | 6.3% |
| | Training place | 9.4% |
| | Training time | 7.8% |
| | Training effect | 7.8% |
| | Others | 4.6% |
| What kind of training do you want to get? | Technical training | 41.8% |
| | Management training | 20.9% |
| | Market training | 17.9% |
| | Sales training | 14.9% |
| | Others | 4.5% |

### 3.3. "Unexpected" Achievements: Multiple Achievements of LTI Development

For decades, LTI has developed vigorously while relying on Nujiang Prefecture's ecological advantages of extensive forestland and a warm climate. By 2022, the planting area of *Lanxangia tsaoko* had reached 74,333 hm$^2$, which accounts for 55.7% of the whole prefecture and 66% of Yunnan province. Additionally, there were 128 enterprises, professional cooperatives, and others engaged in LTI, with an annual processing capacity of 50,000 tons of fresh fruit and an annual production and processing capacity of 9400 tons of dried fruit

in Nujiang Prefecture. More than 90% of *Lanxangia tsaoko*'s primary processing was at the origin, and its output value had reached CNY 1.32 billion (Figure 3). Regarding the increase in farmers' income, the LTI involves 116 village committees in 21 towns of three counties (cities). It covers 43,100 households and 165,000 farmers, including 26,800 households and 82,400 farmers who have been taken out of poverty, which directly drives the per capita annual income increase of about CNY 2000 [54]. The contribution rate of LTI income to the villagers' per capita income is more than 70%, which helps reduce poverty and realize rural revitalization. Nujiang Prefecture is the core main production area of *lanxangia tsaoko* in China. Moreover, *lanxangia tsaoko* is the dominant industry with the strongest driving force, the broadest radiation, and the largest contribution rate in Nujiang Prefecture, and it is an important industry to consolidate the poverty alleviation achievements and effectively achieve rural revitalization.

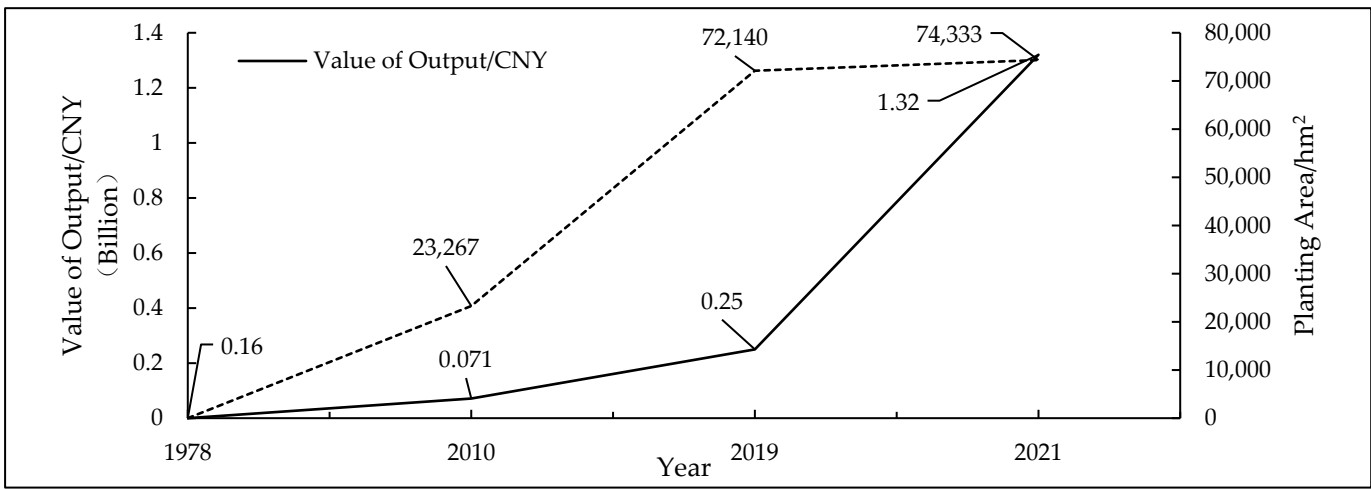

**Figure 3.** Planting area (dashed line) and output value (solid line) of *lanxangia tsaoko* in Nujiang Prefecture from 1978 to 2021.

Apart from the obvious achievements in economic benefits and improving farmers' livelihoods, the development of LTI has effectively promoted infrastructure construction in forest areas, as well as improved the ecological environment of human settlements. On the one hand, the infrastructure conditions in the forest areas have been greatly improved. For example, the rates of access to electricity, water, and highways in the forest areas have reached 95%, 80%, and 85%, respectively. On the other hand, the development of LTI also achieves remarkable ecological benefits. The development of LTI enables to make full use of its advantages and avoid its disadvantages in Nujiang Prefecture. Since 1978, the forest coverage rate of Nujiang Prefecture has reached 78.9%, which is higher than the average level of Yunnan Province. In addition, the development of LTI in Nujiang Prefecture has also promoted the improvement of rural human settlements. By 2021, the treatment of domestic wastewater in Nujiang Prefecture has reached 85.5%, and the treatment of production and domestic garbage has also reached 70.6%. At the same time, 86.5% of households were satisfied with the ecological environment.

*3.4. Development Mode of LTI*

In the process of industrial development, the main factors of production are land, capital, technology, labor, information, and management. Under the relevant roles of stakeholders, including government, enterprises, cooperatives, and farmers, LTI has developed rapidly in Nujiang Prefecture. As shown in Figure 4, the production factors in industrial development have been effectively guaranteed. There are two main development modes of LTI in Nujiang Prefecture. Mode A is related to "enterprises + e-commerce platform + farmers", which means that the three parties cooperate to develop LTI. In this model, enterprises can provide technology and information for industrial development, and e-

commerce platforms will provide channels for product sales. Accordingly, after solving the technical and market problems, farmers are more willing to provide sufficient forestland and participate in LTI. Mode B is related to "cooperatives + primary party organizations + farmers". Similarly, cooperation among the stakeholders will also promote LTI. As the most important organizational form in rural development, cooperatives and primary party organizations can provide some technical and management services for LTI.

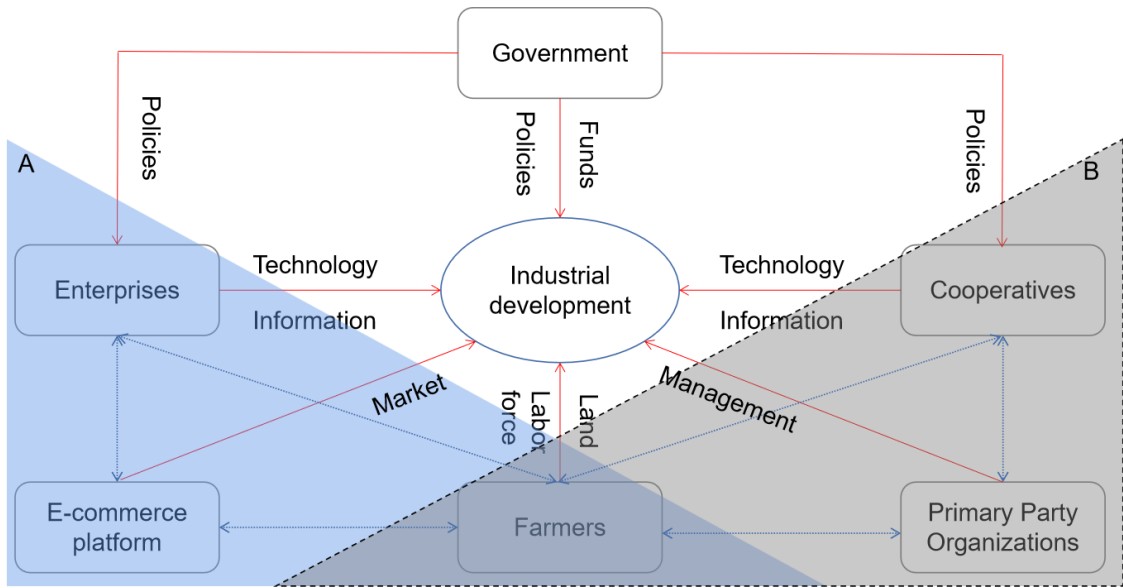

**Figure 4.** Development modes of LTI in Nujiang Prefecture. Notes: A or B refers to two different modes of industrial development. The blue two-way arrow refers to the correlation between different subjects. The red one-way arrow refers to the types of factors of production that different subjects provide to industrial development.

First, some policies can effectively make up for market failure, promote the rationalization and upgrading of industrial structure, and realize the optimal allocation of industrial resources. Markets can better realize the effective allocation of resources, but they are often blind. Meanwhile, their role is mainly to adjust afterwards, inevitably leading to a waste of resources. According to scientific foresight, policies can be adjusted to avoid unnecessary idle resources in advance. Therefore, in order to promote the development of LTI and enhance the industrial competitiveness, the Nujiang Prefecture government issued a series of policies (Appendix A Table A1).

Second, essential funds can strengthen the infrastructure construction and enhance the enthusiasm of farmers, so as to promote industrial sustainable development. As shown in Table 9, the government has successively integrated a large amount of funds for supporting the development of the whole industry chain of *Lanxangia tsaoko* in Nujiang Prefecture.

Third, industrial organizations can promote professional cooperation among enterprises and improve product market competitiveness. The government introduced and fostered service providers for the purchase, processing, and sales of *Lanxangia tsaoko* to improve the organizational level of the development of LTI. On the one hand, the company (Yunnan Energy Investment of Nujiang Industrial Development and Investment Co., Ltd., which is located in Lushui City, Nujiang Prefecture, Yunnan Province) explored the operation mode of "company + farmers + e-commerce platform". On the other hand, the government established professional cooperatives by integrating land, labor, capital, and technology, so as to realize efficient development, and drive the whole prefecture's LTI. For example, there are 36 professional cooperatives in Nujiang Prefecture, which involve 1684 farmers. These cooperatives have achieved an annual sales income of CNY 58.75 million.

**Table 9.** Funds for supporting the development of *Lanxangia tsaoko*.

| Type of Fund | Funds Amount/ Million | Competent Department |
|---|---|---|
| Funds for industrial clustering | CNY 12.4 | Ministry of Agriculture and Rural Affairs |
| Funds for improving quality and efficiency of *Lanxangia tsaoko* | CNY 120 | Nujiang Prefecture Government |
| Funds for scientific and technological innovation and research application for *Lanxangia tsaoko* | CNY 20 | Nujiang Prefecture Government |
| Funds related to forestry industry | CNY 76.46 | Nujiang Prefecture Government |
| Designated assistance fund of China Communications Construction | CNY 10 | China Communications Construction |
| Social capital | CNY 117 | - |

Technology is the foundation and an important guarantee of industrial development. At present, there are three organic Green Certification Bases of LTI in Nujiang Prefecture, with an area of 786.7 hm$^2$ and an output of 3,900,000 kg of products. At the same time, most farmers have received training on the development of LTI, which will provide a strong guarantee for the sustainable development of LTI.

Finally, farmers gradually realize the benefits from the development of LTI, and their enthusiasm for LTI is increasing. The questionnaire survey revealed that 79.6% of farmers were willing to expand the scale of LTI, and 38.7% of farmers were very optimistic about its development.

## 4. Discussion

### 4.1. Experiences of Parallel Chain of "Visible" and "Invisible"

Through the analysis of the development stage of LTI in Nujiang Prefecture, it can be seen that there actually exist in parallel an external promotion chain and an internal breakthrough chain within it. On the one hand, with the external support of government policies and capital, the development of LTI has been strongly guaranteed. On the other hand, under the endogenous incentive of farmers, LTI has achieved remarkable development. As shown in Figure 5, under the interweaving and interaction chains, there is an endogenous force which has promoted LTI development.

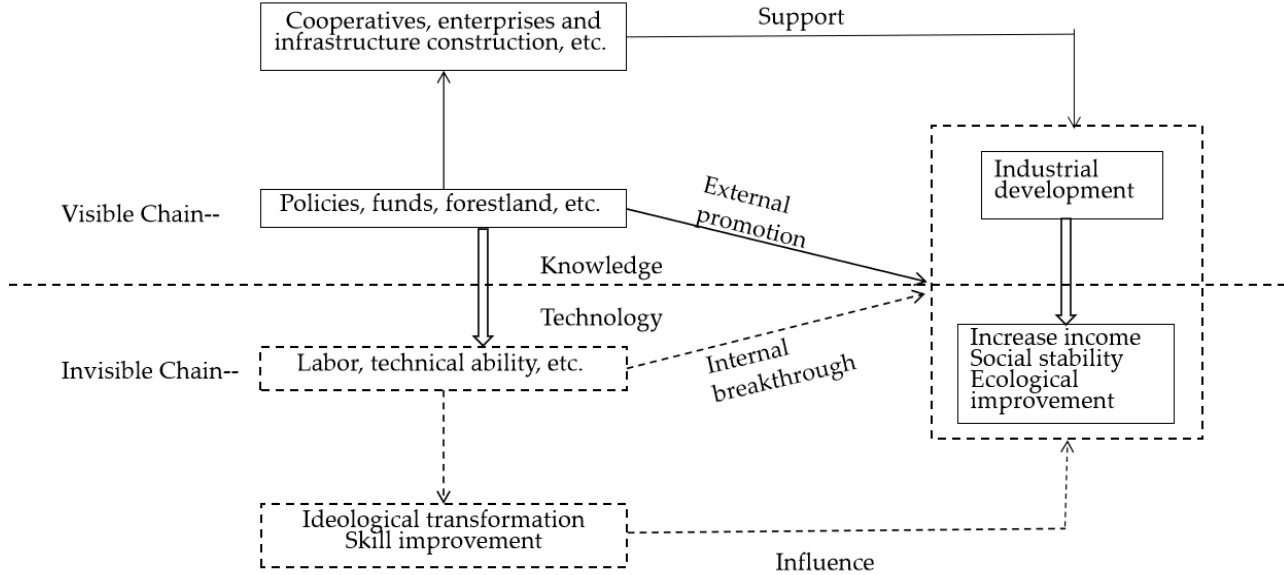

**Figure 5.** Development mechanism of LTI in Nujiang Prefecture.

With the support of government policies and funds, the visible chain is to provide forestland, capital, and organizational guarantees for industrial development by improving the infrastructure and encouraging the participation of cooperatives, enterprises, and other new business organizations. For decades, the government has successively formulated

many policies (Appendix A Table A1) and given financial support to guide industrial development scientifically, reasonably, in an orderly manner, and moderately. The external force is not only reflected in the policy and funding, but also in the improvement of infrastructure construction and innovative organization management under the guidance of the policy.

Obviously, some resourceful farmers can rely on the development of LTI to achieve poverty alleviation. Under the influence of capable farmers, more and more farms realize the development prospects of LTI. Meanwhile, this paper found that 79.6% of the surveyed farmers were willing to expand the scale of the industry. Furthermore, the government or enterprises will give support to farmers in terms of knowledge and technology, which will improve the industrial development ability of farmers. The results show that 18.4% of respondents have received technical support from enterprises, and 27.4% have received scientific and technological services from cooperatives, which have met the development needs of LTI. Thus, the transformation of farmers' ideas and the upgrading of their skills can promote the sustainable development of the industry.

In conclusion, the external promotion chain can help some farmers, who have positive ideas and outstanding abilities, leave poverty. In this process, it also provides farmers with knowledge and technology, and it mobilizes other farmers' awareness of active development, which is the formation process of the internal breakthrough chain. Thus, the interwoven interaction of two chains provides the industry with endogenous development momentum, which is also key to the success of LTI.

*4.2. Mechanism of LTI on Farmers' Livelihood*

Based on industrial development practice, this paper found that the mechanism of LTI on farmers' livelihood is the joint action of government, enterprises, primary party organizations, and farmers. Based on their behavioral motivations, they participate in the industry with their land, capital, labor, information, services, and other production factors. Additionally, the stakeholders will use their advantages to jointly promote the comprehensive role of production factors, so as to promote farmers' self-development ability and achieve the objectives of each entity. Finally, it will promote the overall goal of sustainable development of social economy (Figure 6).

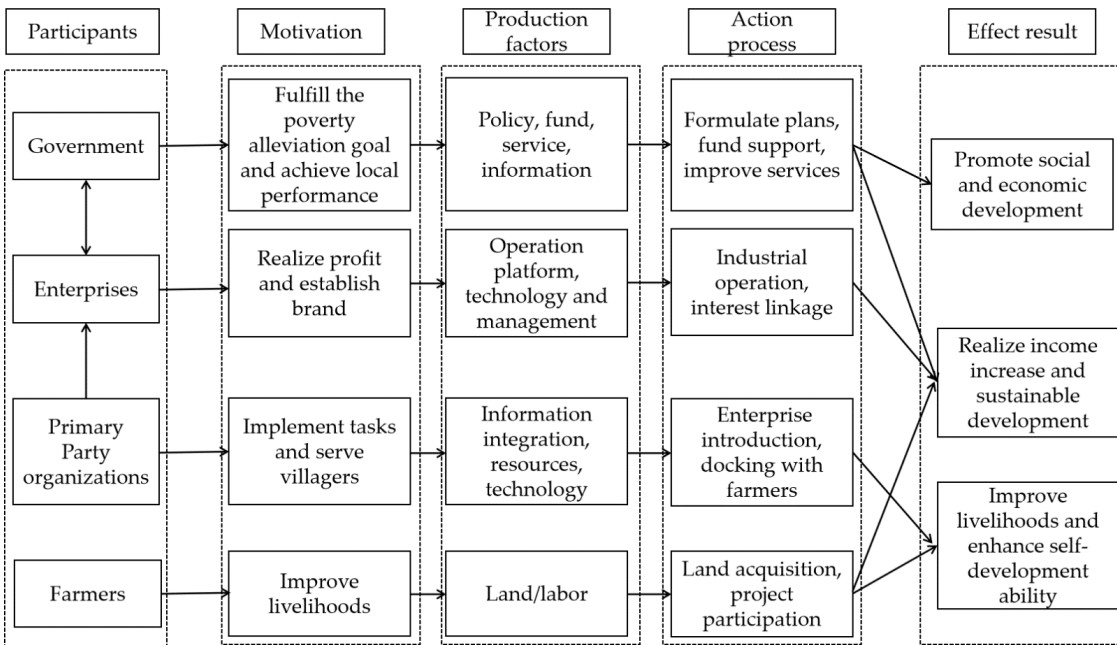

**Figure 6.** Mechanism of industrial development on farmers' livelihoods.

*4.3. Constraints on the Development of LTI*

Some papers say that the forestland area and the degree of forestland fragmentation can affect the enthusiasm of farmers for forestry production [18]. In our survey, it was found that the problem of forestland fragmentation is serious because of various factors, such as forestland equally contracted to households, its intergenerational inheritance, and pressure of population growth in Nujiang Prefecture. Additionally, 3.2% of farmers lack forestland. Both are natural resource factors restricting the development of LTI. Many studies have shown that capital, technology, and organization are vital factors for the development of the forestry industry [55]. For example, it was found that the lack of knowledge and experience, limited institutional capacity, and lack of funding have restricted the wide implementation of agroforestry systems in Timor Leste [56]. Similarly, 28.4% of respondents thought that there are financial problems in the development of LTI, and 23.2% of them believed that technical problems are also the main problems in Nujiang Prefecture. As for the organization and management of industrial development, the main forms of organization are enterprises and cooperatives in Nujiang Prefecture. Enterprises and cooperatives have promoted the development of LTI; however, some respondents did not know enterprises (55.1%) and cooperatives (10.2%) in the local area. The support and services provided by enterprises or cooperatives are limited, which cannot guarantee the further development of LTI. Additionally, the low quality of human capital is key to the lack of sustainability of industrial development [57]. Like most rural areas, the problem of "hollowing out" in Nujiang Prefecture is serious [58], which means that the young labor force mostly chooses to leave the prefecture to work, while the suitable labor force among the elderly and children left behind is smaller. We found that only 37.6% of farmers were between 16 and 40 years old, while 30.6% of those who participated in LTI were over 60 years old. The planting and collection of *Lanxangia tsaoko* are highly technical, and thus it is necessary that farmers receive special training to ensure its sustainable development. However, 40.8% of respondents have not participated in the relevant training, and 10.2% of those who have participated in the training were not satisfied. Therefore, the lack of appropriate labor force and training are also the main factors limiting the development of LTI in Nujiang Prefecture.

## 5. Conclusions

In this paper, we took Nujiang Prefecture as a typical case and analyzed the achievements of LTI and its successful experience using a questionnaires survey and field observation. By exploring the development experience of LTI, we found the following: (1) The LTI development not only improves the income level of farmers, but also improves the living environment and the living standards of farmers in Nujiang Prefecture. (2) There are two main development modes of LTI in Nujiang Prefecture. One involves "enterprises + e-commerce platform + farmers" while the other involves "cooperatives + primary party organizations + farmers". (3) With the interaction of external promotion and internal breakthroughs, the development of LTI has made remarkable achievements. (4) The mechanism of LTI on farmers' livelihoods is the joint action of government, enterprises, primary party organizations, and farmers. Additionally, through the questionnaire survey, we found that (1) Respondents are willing to participate in the development of LTI and are optimistic about the development prospects of LTI. (2) Cooperatives or enterprises provide necessary help and support for farmers to develop LTI. (3) The development of LTI still faces the constraints of forestland, capital, labor, technology, and organization, etc. Finally, based on our analysis, in order to enhance the role of agroforestry in improving farmers' livelihoods, some low-income forest areas can do the following: (1) The government should introduce policies to ensure the development of LTI, and provide necessary funds for LTI. (2) Enterprises, cooperatives, and other organizations provide financial and technical support for the development of LTI, and provide diversified channels for product sales. (3) Farmers should actively participate in LTI and provide forestland and labor force for LTI.

**Author Contributions:** Conceptualization, Y.D. and X.C.; methodology, Y.D. and X.Z.; validation, Y.D. and Y.L.; formal analysis, Y.D.; investigation, Y.D. and Y.L.; resources, X.Z.; data curation, Y.L.; writing—original draft preparation, Y.D. and Y.L.; writing—review and editing, M.L. and X.Z.; supervision, X.C. All authors have read and agreed to the published version of the manuscript.

**Funding:** This research was funded by the 14th Five-Year Plan Pioneering Project of High Technology Plan of the National Department of Technology under Grant, grant number 2021YFD2200405.

**Institutional Review Board Statement:** Not applicable.

**Informed Consent Statement:** Informed consent was obtained from all the individual participants included in the study.

**Data Availability Statement:** The data presented in this study are available on request from the corresponding author.

**Acknowledgments:** All authors gratefully acknowledge the support of the People's Government of Nujiang Lisu Autonomous Prefecture that participated in the investigation, especially the Forestry and Grassland Administration for processing and providing the detailed survey data.

**Conflicts of Interest:** The authors declare no conflict of interest.

## Appendix A

**Table A1.** Policies of Nujiang Prefecture to promote the development of LTI.

| Policy | Goal | Organization | Date |
|---|---|---|---|
| Opinions on Accelerating the Development of LTI | Putting the LTI in the prominent position of ecological construction and rural economic development, promoting the development of LTI, and overcoming poverty. | Rural and Agriculture Administration of Nujiang Lisu Autonomous Prefecture | July 2014 |
| Management Measures for the Development of LTI in Nujiang Prefecture | Strengthening and standardizing the selection and breeding of *Lanxangia tsaoko*, promoting the industrialization of *Lanxangia tsaoko*, and promoting its sustainable development. | People's Government of Nujiang Lisu Autonomous Prefecture | January 2016 |
| Nujiang Prefecture Green Spice Industry Construction Plan (2018–2022) | Building Nujiang Prefecture into a green spice industry cluster with *Lanxangia tsaoko* as the core, which is leading development level and significant comprehensive benefits. | People's Government of Nujiang Lisu Autonomous Prefecture | April 2018 |
| Three Year Action Plan for Improving Quality and Efficiency of *Lanxangia tsaoko* in Nujiang Prefecture (2022–2024) | Achieving the goal of doubling the output and output value of *Lanxangia tsaoko* in Nujiang Prefecture, and improving the processing and transformation capacity. | People's Government of Nujiang Lisu Autonomous Prefecture | April 2022 |

**Table A2.** Willingness and behavior of farmers in LTI.

| Questionnaire Content | Basic Questions | Option Settings |
|---|---|---|
| Basic Information | Gender | Male    Female |
| | Age | 16–30 years old |
| | | 31–40 years old |
| | | 41–60 years old |
| | | Over 60 years old |
| | Nationality | Han |
| | | Lisu |
| | | Other |
| | Education | Not attending school |
| | | Primary (Grade 1–6) |
| | | Middle (Grade 7–9) |
| | | High (Grade 10–) |
| | Social security | Medical insurance |
| | | Endowment insurance |

**Table A2.** *Cont.*

| Questionnaire Content | Basic Questions | Option Settings |
|---|---|---|
| Willingness of the surveyed farmers to choose LTI | Are you willing to expand industry scale? | Yes    No |
| | The attitude of industrial development | Very optimistic |
| | | Quite optimistic |
| | | Average |
| | | Quite pessimistic |
| | | Very pessimistic |
| | The problem of industrial development | Lack of capital |
| | | Lack of technology |
| | | Lack of labor |
| | | Lack of sales channels |
| | | Lack of infrastructure |
| | | Lack of resources |
| | | Lack of projects |
| | What kind of support do you hope to get? | Financial support |
| | | Technical support |
| | | Talent support |
| | | Market support |
| | | Information support |
| | | Policy support |
| | The form of industrial organization | Farmers themselves |
| | | Enterprises |
| | | Village collectives |
| | | Cooperatives |
| | | Government |
| Willingness of the surveyed farmers to choose organizational form | Whether you joined cooperatives | Yes |
| | | No |
| | | Unclear |
| | What services have you received from cooperatives? | Production services |
| | | Technology services |
| | | Transportation and marketing services |
| | | Processing services |
| | | Loan services |
| | | No specific services |
| | What kind of service do you want to get? | Financial services |
| | | Technical guidance |
| | | Product sales |
| | | Talent training |
| | | Market support |
| | | Information services |
| | Your satisfaction with cooperatives | Very satisfied |
| | | Quite satisfied |
| | | Average |
| | | Quite unsatisfied |
| | | Very unsatisfied |
| | Whether you joined enterprises | Yes |
| | | No |
| | | Unclear |
| | What support have you received from enterprises? | Financial support |
| | | Technical support |
| | | Talent support |
| | | Market support |
| | | Information support |
| | | No specific support |
| | What kind of support do you want to get? | Financial support |
| | | Technical guidance |
| | | Product sales |
| | | Talent training |
| | | Market support |
| | | Information services |
| | Management mode of enterprises cooperation | Enterprises + farmers |
| | | Enterprises + bases + farmers |
| | | Enterprises + cooperatives + farmers |
| | | Others |

**Table A2.** *Cont.*

| Questionnaire Content | Basic Questions | Option Settings |
|---|---|---|
| Willingness of the surveyed farmers to choose training | Are you willing to participate in training? | Yes<br>No |
| | Have you participated in training? | Yes<br>No |
| | Please evaluate the training effect | Very satisfied<br>Quite satisfied<br>Average<br>Quite unsatisfied<br>Very unsatisfied |
| | Factors influencing farmers' participation in training | Personal concern<br>Training content<br>Training methods<br>Training place<br>Training time<br>Training effect<br>Others |
| | What kind of training do you want to get? | Technical training<br>Management training<br>Market training<br>Sales training<br>Others |

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
