# Peer review of "The Role of Agroforestry in Poverty Alleviation: A Case Study from Nujiang Prefecture, Southwestern China"

_sustainability, doi:10.3390/su151512090_

Round 1

Reviewer 1 Report

Dear editor

In the present study, the researchers investigated the role of agroforestry on poverty alleviation in China based on the SDGs. Specifically, in this research, the researchers have analyzed the impact of agroforestry on farmers' livelihoods and its mechanism using literature, field observation, questionnaires in Nujiang Prefecture.

The present research provides valuable information on the impact of agroforestry on poverty reduction. However, a more comprehensive literature review is needed in the introduction section.

It is also described in a very vague and general way in the methodology section, especially the statistical analysis.

In addition, the following comments should also be corrected

Line 16: A. tsaoko instead of Amomum tsaoko

Lines 21 to 33: It is better to add quantitative (numerical) results to this section.

Line 139: The quality of figure 1 is very low and should be improved. Also, the north direction should be added to the maps.

Line 162: Add the population statistics of each city. Amomum tsaoko planting area should also be expressed as a percentage.

Line 204: Questionnaire information should be provided, including the number of questions and their grouping, as well as the number of questionnaires. Also, a sample questionnaire should be provided in the attached section.

Lines 215 to 223: How to analyze the questionnaire should be explained in more detail. Also, the analyzes used, and the software used should be specified. The level of validity and reliability of the questionnaire results should be analyzed and reported.

Minor editing of English language required

Author Response

We are very grateful to your comments for the manuscript. Your comments and suggestions will play an important role in improving the quality of the manuscript. I have revised and improved this paper in strict accordance with your comments. All of your questions were answered one by one.

Comments:

Dear editor

In the present study, the researchers investigated the role of agroforestry on poverty alleviation in China based on the SDGs. Specifically, in this research, the researchers have analyzed the impact of agroforestry on farmers' livelihoods and its mechanism using literature, field observation, questionnaires in Nujiang Prefecture. The present research provides valuable information on the impact of agroforestry on poverty reduction.

Point 1: However, a more comprehensive literature review is needed in the introduction section.

Response 1: Thank for your the suggestion. We have added literature review in the introduction section.

We have added it as explained above (Lines 55-66, 67-84, page 2).

Point 2: It is also described in a very vague and general way in the methodology section, especially the statistical analysis.

Response 2: Thank for your the suggestion. We have refined the methodology section of the paper. Also, We have increased the statistical analysis. The details are as follows.

2.3.2 Descriptive Statistical Analysis

Descriptive Statistical Analysis is a method that use tabulation and classification, graphs, and calculation of generalized data to describe the characteristics of data. It is a data statistical description of all variables in the survey, including Frequency Analysis, Central Trend Analysis, Dispersion Degree Analysis and some basic statistical graphs. Frequency Analysis is used to calculate the selection frequency and proportion of certain types of data, such as gender. It is used for the statistics of basic background information of samples, as well as the analysis of sample characteristics and basic attitudes. This paper uses SPSS Statistics 29.0 to conduct Frequency Analysis of the questionnaire. 

Point 3: Line 16: A. tsaoko instead of Amomum tsaoko

Response 3:  Thank for your the suggestion. We have revised Amomum tsaoko to A. tsaoko in paper.

Point 4: Lines 21 to 33: It is better to add quantitative (numerical) results to this section.

Response 4: Thank you very much for highlighting this flaw. According to your comments, we have added the statistical analysis results of the questionnaire in the Results.

We have added it as explained above (Lines 292-363, page 7-11).

Point 5: Line 139: The quality of figure 1 is very low and should be improved. Also, the north direction should be added to the maps.

Response 5: Thank you very much for highlighting this flaw. We have refined Figure 1 more clearly. Also, we have added the north direction in the map.

Point 6: Line 162: Add the population statistics of each city. Amomum tsaoko planting area should also be expressed as a percentage.

Response 6: Thank for your the suggestion. We have added the population statistics of each city in Table 1. And we have added the Proportion of A. tsaoko plantingin Table 2.

Point 7: Line 204: Questionnaire information should be provided, including the number of questions and their grouping, as well as the number of questionnaires. Also, a sample questionnaire should be provided in the attached section.

Response 8: Thank you for underlining this deficiency. We have added the questionnaire information in Result section (lines 292-363, page 7-11), And we provided a sample questionnaire in Appendix (Table A2) .

Point 8: Lines 215 to 223: How to analyze the questionnaire should be explained in more detail. Also, the analyzes used, and the software used should be specified. The level of validity and reliability of the questionnaire results should be analyzed and reported.

Response 8: Thank you for underlining this deficiency. This paper uses SPSS Statistics 29.0 to conduct Frequency Analysis of the questionnaire.

By analyzing the reliability and validity of the questionnaire, we find α = 0.826 > 0.7, which shows that the reliability of the questionnaire is very good (Table 4). As shown in Table 5, the value of KMO is 0.876 > 0.8, which indicates that the questionnaire data is very effective.

Reviewer 2 Report

Secondly, the government also should improve infrastructure and industrial 496 park construction, to improve industrial competitiveness as well as promote industrial 497 transformation and upgrading. Thirdly, it is necessary to innovate the organizational 498 mode, then actively introduce new cooperative organizations such as cooperatives and 499 leading enterprises, which can give farmers some support in technology and information. 500 Finally, based on industrial development and farmers' willingness, skills training should 501 be carried out to improve farmers' development ability. What’s more, farmers should 502 change their thinking and actively develop ATI.

 Rest things are ok 

Author Response

Dear reviewer,

We are very grateful to your reviewing for the manuscript. We are sorry that we did not find any comments from you on the manuscript. However, we have revised and improved the paper according to the comments of other reviewers.

Firstly, we have added literature review in the introduction section and the statistical analysis results of the questionnaire in the Results.

Then, we have refined the methodology section of the paper. On the one hand, we have refined the Data Sources in lines 212-233. On the other hand, we have increased the statistical analysis in lines 282-290.

Finally,  we have reorganized the contents of the manuscript. We have refined the content. We have adjusted the development stage of ATI to the Materials and Methods section (lines 160-206, page 4). Also, we have refined the Results and the Discussion.

Thank you again for your review of the manuscript.

Please feel free to contact me if you have any questions.

Reviewer 3 Report

Organization of the manuscript should be improved. For example, authors claim both field survey and theoretical deduction has been used. However, the methodology used is not at all well-organized. This way there are issues in the Results section, which may belong to the Material and Methods section. Also issues in the Discussion and the results section are intertwined.

In general the redaction is very lengthy and repetitive.

Attached is a pdf with several examples, where amelioration is required. But please, authors should be aware that I have not been exhaustive.

Please revise thoroughly all the sentences and paragraphs.

Electronic Files with full details of experimental procedures, data collected and data handling should be included as supplementary material.

The English Language should be very much improved.

A number of sentences are incomprehensible, flaw or even nonsense.

In the attached pdf, again there are several examples. And again,  please, authors should be aware that I have not been exhaustive.

Author Response

We are very grateful to your comments for the manuscript. Your comments and suggestions will play an important role in improving the quality of the manuscript. I have revised and improved this paper in strict accordance with your comments. All of your questions were answered one by one.

Comments:

Point 1: Organization of the manuscript should be improved. For example, authors claim both field survey and theoretical deduction has been used. However, the methodology used is not at all well-organized.

Response 1: Thank you for underlining this deficiency. We have refined the methodology section of the paper. Firstly, we have refined the Data Sources in lines 212-233. Also, we have added the statistical analysis in lines 282-290.

Point 2: This way there are issues in the Results section, which may belong to the Material and Methods section. Also issues in the Discussion and the results section are intertwined.

Response 2: Thank for your the suggestion. We have refined the content. We have adjusted the development stage of ATI to the Materials and Methods section (lines 160-206, page 4). Also, we have refined the Results and the Discussion. You can find our revisions in the revision.

Point 3: Please revise thoroughly all the sentences and paragraphs.

Response 3: Thank for your the suggestion. We have revised all the sentences and paragraphs in paper.

Point 4: Electronic Files with full details of experimental procedures, data collected and data handling should be included as supplementary material.

Response 4: Thank you for your suggestions, and our reply is as follows: firstly, we have added the statistical analysis results of the questionnaire in the Results. Then, we provided a sample questionnaire in Appendix (Table A2) . Also, we have added the data collected in Data sources (lines 216-220).

Thank you again for your comments and suggestions.

We have carefully revised and refined it in the revision.

Round 2

Reviewer 1 Report

Dear editor

The authors have made a lot of effort to improve the article and the quality of the article has increased significantly. Although there are still some minor flaws in its text, their correction before the publication of this article will further improve its quality and increase the number of its readers.

As a proposition that you are free to accept or not, the use of the abbreviation (AF) for the word Agroforestry is not interesting.

Why you used the star sign (*) at the end of the titles of tables 6 and 7?

As I see some grammatical mistake, so the minor editing of English language required.
Despite the modification of Figure 1, its quality is not suitable for publication.

As I see some grammatical mistake, so the minor editing of English language required.

Author Response

Dear reviewer,

Thank you for your careful review. We really appreciate your efforts in reviewing our manuscript during this unprecedented and challenging time. We wish good health to you, your family, and community. Your careful review has helped to make our study clearer and more comprehensive, and our point-by-point responses are presented above. The responses to your comments are marked in red and presented following.

Point 1: As a proposition that you are free to accept or not, the use of the abbreviation (AF) for the word Agroforestry is not interesting.

Response 1:  Thank for your the suggestion. According to your suggestion, we have revised AF to Agroforestry  in the manuscript 

Point 2: Why you used the star sign (*) at the end of the titles of tables 6 and 7?

Response 2:  Thank you for underlining this deficiency. As a result of our error, the star sign (*) at the end of the titles of tables 6 and 7. We've already deleted it.

Point 3:  As I see some grammatical mistake, so the minor editing of English language required.

Response 3:   In order to improve the academic and readable English language of the manuscript, we use the English language service provided by MDPI to polish the manuscript.

Full details of the editing service can be found at https://www.mdpi.com/authors/english.

Point 4: Despite the modification of Figure 1, its quality is not suitable for publication.

Response 4:  We deeply appreciate your suggestion, and we have improved the clarity of Figure 1.

Thank you again for your comments and suggestions.We have carefully revised and refined it in the revision.

Reviewer 2 Report

Dear Sir,

The paper can be accepted 

Author Response

Dear reviewer,

Thank you very much for your affirmation and recognition of the paper. We have revised and improved the paper according to the opinions of other reviewers. You can find our revised version in the attachment.

Thank you again for your review of the manuscript.

Please feel free to contact me if you have any questions.

Reviewer 3 Report

The revised version of the manuscript has been very much improved. However, the manuscript needs very much tightening up, before it can be accepted for publication. Redaction remains a problem as a lot of sentences remain flaw. Still, there are a number of incongruous or incoherent statements.

A main flaw is the inconsistence between objectives and conclusions. First, objectives are more understandable than in the previous version, but still need amelioration. Second, conclusions should be new rewritten. This is because conclusions contain sentences about subjects that now are included in the methodology. This is the case of the four stages recognized for ATI; it is very much inconsistent to present methods as conclusions. Conclusions should be in line with objectives and hypothesis.

Also authors should clarify if policy (or even political) recommendations are and objective.

The English Language should be very much improved to allow or increase readability. Please, authors should be aware that it is boring to record all the language mistakes in this manuscript.

Again, please edit the manuscipt, allowing a correct use of the English Language.

Author Response

Dear reviewer,

We are very grateful to your comments for the manuscript. Your comments and suggestions will play an important role in improving the quality of the manuscript. I have revised and improved this paper in strict accordance with your comments.  All of your questions were answered one by one.

First, in order to improve the academic and readable English language of the manuscript, we use the English language service provided by MDPI to polish the manuscript.

Full details of the editing service can be found at

â–º https://www.mdpi.com/authors/english.

Then,  based on the objectives and hypothesis of the study,we have rewritten the conclusions. These are as follows:

In this paper, we took Nujiang Prefecture as a typical case and analyzed the achievements of ATI and its successful experience using a questionnaires survey and field observation. By exploring the development experience of ATI, we found the following: (1) The ATI development not only improves the income level of farmers, but also improves the living environment of farmers and the living standards of farmers in Nujiang Prefecture. (2) There are two main development modes of ATI in Nujiang Prefecture. One is about “enterprises + e-commerce platform + farmers”. Another is about “co-operatives + primary party organizations + farmers”. (3) With the interaction of external promotion and internal breakthroughs, the development of ATI has achieved remarkable achievements. (4) The mechanism of ATI on farmers' livelihood is the joint action of government, enterprises, primary party organizations, and farmers. Additionally, through the questionnaire survey, we found that (1) Respondents are willing to participate in the development of ATI and are optimistic about the development prospects of ATI. (2) Co-operatives or enterprises provide necessary help and support for farmers to develop ATI. (3) The development of ATI still faces the constraints of forestland, capital, labor, technology and organization, etc. Finally, based on our analysis, in order to enhance the role of agroforestry in improving farmers' livelihood, some low-income forest areas can do some things from the following aspects: (1) The government should introduce policies to ensure the development of ATI, and provide necessary funds for ATI. (2) Enterprises, co-operatives and other organizations provide financial and technical support for the development of ATI, and provide diversified channels for product sales. (3) Farmers should actively participate in ATI and provide forest land and labor force for ATI.

Also, we have refined the Abstract and Methods in the manuscript.

Thank you again for your comments and suggestions.  We have carefully revised and refined it in the revision. You can find our specific modifications in the attachment.  Please feel free to contact me if you have any questions.

Round 3

Reviewer 3 Report

The conslusions have been somewhat improved.

However, authors have not been been able to answer all the sent questions.

In my opinion this behaviour is inappropriate, but editors should decide.

Still a numbe rof flaws, but editors decide.